# In Vivo Monitoring of Acetylcholine Release from Nerve Endings in Salivary Gland

**DOI:** 10.3390/biology10050351

**Published:** 2021-04-21

**Authors:** Masanobu Yoshikawa, Mitsuru Kawaguchi

**Affiliations:** 1Department of Clinical Pharmacology, School of Medicine, Tokai University, Isehara 259-1193, Japan; 2Tokyo Dental College, Tokyo 101-0061, Japan; kawaguti@tdc.ac.jp

**Keywords:** salivary gland, acetylcholine, microdialysis

## Abstract

**Simple Summary:**

Stimulation of the parasympathetic nervous system results in the secretion of saliva. Previous studies have demonstrated acetylcholine content in homogenate obtained from salivary glands. Acetylcholine in homogenate, however, includes that stored in the cells, as well as that released in the interstitial fluid. The activity of the parasympathetic nervous system is mainly determined by the amount of acetylcholine released. We established an in vivo microdialysis method for monitoring the acetylcholine released from nerve endings in the salivary glands in rats. The results of the present study demonstrated that acetylcholine levels in the dialysate reflect acetylcholine levels in the interstitial fluid of the submandibular gland, and that an increase in the acetylcholine level in the dialysate depends predominantly on the release of acetylcholine from the parasympathetic nerve endings.

**Abstract:**

A microdialysis technique was used to monitor acetylcholine levels in the local interstitial fluid in rat submandibular glands, with the aim of determining parasympathetic nerve activity in vivo. The dialysis probe housed a 10 × 0.22 mm semipermeable membrane (molecular weight cutoffs: 50,000 Da). When the probe was perfused at 2 μL/min in vitro, the mean relative recovery of acetylcholine was 41.7% ± 2.5%. The dialysis probes were implanted in the submandibular glands of anesthetized rats and perfusion with Ringer’s solution, at 2 μL/min, was performed. Acetylcholine concentrations in the dialysate were measured by high-performance liquid chromatography and electrochemical detection. The results revealed the following: (1) that mixing Eserine with Ringer’s solution allowed acetylcholine in the salivary glands to be quantified; (2) that acetylcholine concentrations in the dialysate were highly variable and unstable over the first 120 min after probe implantation, but reached a nearly stable level (4.8 ± 2.7 nM) thereafter in the presence of 100 µM of Eserine; and (3) that electrical stimulation of the *chorda tympani* nerve, or perfusion with high potassium Ringer’s solution, significantly increased acetylcholine concentrations in the dialysate. These results indicate that the present microdialysis technique offers a powerful tool for detecting changes in parasympathetic activity within the salivary glands.

## 1. Introduction

In contrast to most organs, where the parasympathetic and sympathetic nerves act antagonistically, both types of nerve act in coordination to regulate the salivary glands. Parasympathetic excitation causes the nerve endings to release acetylcholine, a neurotransmitter that acts on the salivary glands. The action of this neurotransmitter on the salivary glands primarily stimulates the secretion of water and ions [1,2,3]. Sympathetic excitation, on the other hand, leads to the release of noradrenaline from postganglionic nerve endings. Noradrenaline acts on the salivary glands to stimulate the secretion of proteins [4,5,6,7].

The stimulation of the parasympathetic nervous system results in the secretion of large amounts of saliva due to the binding of acetylcholine to muscarinic acetylcholine receptors, mainly those for M3 receptors, on acinar cells [3,8]. The muscarinic acetylcholine receptors activate phospholipase C, which hydrolyzes phosphatidylinositol 4,5-bisphosphate into the inositol 1,4,5-trisphosphate and diacylglycerol. This elevates the concentration of cytoplasmic calcium released from storage in the endoplasmic reticulum, leading to the activation of ion transporters and ion channels.

Previous studies have demonstrated the neurotransmitter content in homogenate obtained from rat and mouse salivary glands [9,10]. The chronic administration of isoprenaline, a non-selective β-adrenergic agonist, or streptozotocin, a diabetogenic drug, altered both the amount of saliva and the acetylcholine content in homogenate obtained from salivary glands [11,12]. These studies suggest that change in acetylcholine levels in homogenate from salivary glands results in the modification of salivary secretory function. Acetylcholine in homogenate, however, includes that stored in the cells, as well as that released in the interstitial fluid. The activity of the parasympathetic nervous system is mainly determined by the amount of acetylcholine released. This is supported by the results of earlier studies in which the salivary glands were perfused with parasympathomimetics [13,14,15,16], or in which electrical stimulation of the nerves resulted in a significant increase in salivary flow [17]. This indicates that it is necessary to measure the amount of acetylcholine released in order to determine parasympathetic activity in the salivary glands.

In vivo microdialysis has mainly evolved in the field of brain science as it allows the collection of released neurotransmitters in the synapses through a semipermeable membrane [18,19]. The results of such microdialysis analyses are thought to reflect the physiological and functional significance of regional neurotransmitters. This technique is applicable to peripheral organs as well [20,21,22,23]. The application of in vivo microdialysis to the salivary glands may offer potential advantages over conventional methods using tissue homogenates because it allows continuous long-term sampling of the interstitial solutes within a defined area of the salivary gland in which the dialysis probe has been implanted. The purpose of the present study was to evaluate parasympathetic activity in the salivary glands under various physiological conditions in vivo by using a microdialysis method to monitor the acetylcholine released from nerve endings in the salivary glands of anesthetized rats.

## 2. Materials and Methods

All the animal experiments in the present study were performed strictly in strict accordance with the Guidelines for the Care and Use of Animals for Scientific Purposes at Tokai University (https://www.u-tokai.ac.jp/about/compliance/animal-experiments/guidance/). Approval for the study protocol was obtained from the Animal Investigation Committee of Tokai University (Approval No: 171045, 182019 and 193029).

### 2.1. Animals

Male Wistar rats (8–9 weeks old, 230–280 g each, *n* = 18; Nihon Clea, Tokyo, Japan) were housed in an air-conditioned room at a control temperature of 24–26 °C and 50–60% humidity, with a 12-h light/dark cycle (lights on: 07:00, and food and water freely available. The animals were allowed 1 week to adapt to the novel laboratory environment.

### 2.2. Chemicals

The following were obtained from the sources indicated: isopropylhomocholine (IPHC; Eicom, Kyoto, Japan) and Eserine (physostigmine; Tokyo Chemical Industry Co., Tokyo, Japan). Unless otherwise indicated, all chemicals were purchased from Nacalai Tesque Japan (Kyoto, Japan).

### 2.3. Microdialysis in Anesthetized Rats

A linear dialysis probe (OP-100-10, Eicom, Kyoto, Japan) was implanted in the submandibular glands of rats under inhalation anesthesia with nitrous oxide, oxygen, and isoflurane (2%). The semipermeable membrane region (10 mm) of the probe was implanted in the left submandibular gland along the long axis (Figure 1). The probes were perfused with Ringer’s solution at a speed of 2 µL/min by a micro-infusion pump (ESP-32, Eicom, Kyoto, Japan). The Ringer’s solution consisted of 147 mM NaCl, 2.2 mM CaCl_2_, 4.02 mM KCl, and the cholinesterase inhibitor Eserine (physostigmine; 100 µM). To elicit the release of acetylcholine from the nerve endings, the microdialysis probe was perfused with Ringer’s solution containing a higher than usual amount of KCI (147 mM NaCl, 2.2 mM CaCl_2_, 100 mM KCl, 100 µM Eserine) (high-K^+^ Ringer’s solution). The perfused solution was switched to high-K^+^ Ringer’s solution and back to regular Ringer’s solution by using an SI-60 liquid switch (Eicom, Kyoto, Japan).

Each sampling period lasted 15 min (sample volume = 30 µL), which allowed for the sufficient collection of effluent for a quantitative determination of acetylcholine level. Each 15 min sample was collected in a chilled microtube containing 3 µL of 3 nM IPHC as an internal standard.

### 2.4. Acethylcholine Determination

The collected solution was subjected to high-performance liquid chromatography (HPLC) with the HTEC-500 (Eicom, Kyoto, Japan) and a platinum electrode (WE-PT, Eicom, Kyoto, Japan). The samples were separated on a polymer-based reverse-phase column (φ 2.0 mm × 150 mm; Eicompak AC-GEL; Eicom, Kyoto, Japan) and then subjected to an enzymatic reaction on an enzyme column (φ 1.0 mm × 4 mm; AC-ENZYM1; Eicom, Kyoto, Japan) at 35 °C and a guard column (for sample, φ 3.0 × 4 mm; PC-03-CH; for mobile phase, PC-04-CH, φ 4.0 × 5 mm). The mobile phase consisted of 5 g/L KHCO_3_ including 50 mg/L ethylenediaminetetraacetic acid disodium (EDTA·2Na) and 300 mg/L sodium 1-decanesulfonate. The electrode potential was set at +450 mV against a Ag/AgCl reference electrode. The quantification of the collected acetylcholine was evaluated by using a peak area ratio relative to that of IPHC as the internal standard. The data were collected and analyzed using the PowerChrome software (eDAQ, Denistone East, Australia). The present method achieved a detection limit of acetylcholine at 0.05 nM (1 fmol) and a quantitation limit of 0.2 nM (4 fmol) at a signal-to-noise ratio of 3. The quantification of the collected acetylcholine was evaluated with the IPHC peak area as the internal standard.

### 2.5. Electrical Stimulation of Parasympathetic Nerve

The *chorda tympani* nerve was exposed and stimulated electrically to induce the release of acetylcholine from the parasympathetic nerve end [17,24]. A pair of stimulation electrodes (stainless wire 0.5 mm in diameter, 5 mm interpolar distance) was settled on the nerve. Square-wave pulses of 5 msec in duration and 1.5–2.0 V in intensity were applied as electrical stimuli at frequencies of 20 Hz using an electronic stimulator SEN-3201 (Nihon Koden, Tokyo, Japan).

### 2.6. Statistical Analyses

The results given represent the mean and standard deviation (SD) of the results. A statistical analysis software package (GraphPad Prism, version 6.0c, GraphPad Software, San Diego, CA, USA) was used to compare across the experimental conditions. Dunn’s multiple comparison test was used to determine significance at each time point when a significant difference among acetylcholine levels after perturbation was obtained by means of a two-way (drugs and time) repeated-measures analysis of variance (ANOVA). A *p*-value of less than 0.05 was considered to indicate a statistical significance.

## 3. Results

### 3.1. Microdialysis Condition

The injection flow rate and analyte concentration were varied in order to optimize the time resolution and to define the sensitivity of the analytical assay and the sample collection volume. The relationship between the perfusion speed and the relative and absolute recovery rates were investigated in vitro to determine an adequate dialysis perfusion speed (Figure 2A). Five dialysis probes were immersed in the testing solution consisting of Ringer’s solution with a constant concentration of acetylcholine (10 nM; testing solution) at 37 °C, and the dialysate samples were collected at various perfusion speeds (1–5 µL/min). The concentrations of acetylcholine in the dialysates obtained from the five different probes were measured by HPLC. Raising the perfusion speed from 1 μL/min to 5 μL/min yielded a decrease in the relative recovery of acetylcholine [(concentration in dialysate)/(concentration in testing solution)]. In contrast, the absolute recovery rate [(concentration in dialysate) × (perfusion speed)] showed a nonlinear increase, approaching an almost steady-state at 2 µL/min. Using this perfusion speed (2 µL/min) in vitro, a nearly uniform relative recovery rate (mean 41.7% ± 2.5%) was obtained, even when the acetylcholine concentration in the testing solution was changed (1–5 nM) (Figure 2B).

### 3.2. Effect of Eserine on Basal Level of Acetylcholine

Where cholinergic mechanisms are implicated, acetylcholinesterase (EC 3.1.1.7) is present in the distribution. No acetylcholine was detected in the dialysate in the absence of Eserine, an acetylcholinesterase inhibitor. Perfusion with various concentrations (1–200 μM) of Eserine through the dialysis membrane increased the basal acetylcholine level in a dose-dependent manner, with an almost steady-state being approached at 100 μM (Figure 3).

### 3.3. Time Course of Dialysate Acetylcholine Levels after Implantation of Probe

Figure 4 shows the time course of the change in acetylcholine level in the dialysate collected at 15-min intervals over a period of 360 min in the presence of 100 µM Eserine. The acetylcholine levels were highly variable and unstable over the first 120 min after implantation of the probe, but showed a gradual decrease to basal level (4.9 ± 2.2 nM) and a stable state (4.8 ± 2.7 nM) thereafter.

### 3.4. Response to Electrical Stimulation of Chorda Tympani Nerve

Figure 5 shows changes in acetylcholine levels in the dialysate due to electrical stimulation of the *chorda tympani* nerve in the presence of 100 µM Eserine. The stimulation significantly increased acetylcholine levels in the dialysate by 2854% ± 804% of the basal level (*p* < 0.05). After stimulation, the acetylcholine levels in the dialysate returned to 91% ± 14% of the basal level.

### 3.5. Response to High-K^+^ Ringer’s Solution

Figure 5 shows changes in acetylcholine levels in the dialysate due to high-K^+^ Ringer’s solution in the presence of 100 µM Eserine. Perfusion with high-K^+^ Ringer’s solution yielded a significant increase in acetylcholine levels in the dialysate by 908% ± 251% of the basal level (*p* < 0.05). The acetylcholine levels in the dialysate returned to 121% ± 91% of the basal level at 15 min after perfusion with high-K^+^ Ringer’s solution, and then decreased by 24% ± 8% at 30 min.

## 4. Discussion

The results of the present study demonstrated that the microdialysis method used allowed specific monitoring of the release of acetylcholine in rat submandibular glands. Furthermore, electrical stimulation of the *chorda tympani* nerve or perfusion with high-K^+^ Ringer’s solution resulted in an increase in acetylcholine levels in the dialysate. Microdialysis techniques have long been used to study the dynamics of neurotransmitters in the brain [18,19]. To the best of our knowledge, this is the first report on the detection of the release of endogenous neurotransmitters in salivary glands in vivo.

Although the role of acetylcholine as a neurotransmitter in the regulation of salivary glands has been known for a long time, it has been difficult to find a specific and sensitive analytical method for measuring it in trace amounts. Previous studies have found acetylcholine in homogenate obtained from salivary glands in rats or mice [9,10]. In the present study, microdialysis and highly sensitive HPLC allowed the release of acetylcholine to be monitored in the salivary glands. These methods enable sequential monitoring of neurotransmitters under a variety of pathophysiological conditions in vivo. For example, while saliva flow showed a marked decrease in streptozotocin-induced diabetic mice, acetylcholine in homogenates obtained from their salivary glands showed a significant increase [12]. Cellular mechanisms, such as the synthesis of acetylcholine and compensatory changes due to diabetes, have failed to explain this discrepancy.

Assuming that the in vivo recovery rate is comparable with the in vitro recovery rate, it may be possible to estimate the interstitial acetylcholine concentrations in rat submandibular glands using the latter (approximately 41%). The estimated means of the values recorded at 0–60 min and during the steady state were 17.3 and 11.7 nM, respectively, in the present study. The estimated initial and steady-state values were approximately 23 and 15 times greater, respectively, than that of plasma (0.75 nM) in another study [25]. These results suggest that the levels of acetylcholine in dialysate correspond to those derived from nerves in the salivary glands. One earlier study found that the concentration of acetylcholine in homogenate from submandibular glands in rats (5 weeks old) was 14.1 (nmol/g tissue) [10]. This value is approximately 1200 times greater than that observed in the present study (11.7 nM). Taken together, these results indicate that the concentration of acetylcholine in the homogenate is mainly due to the portion stored in the nerves. Monitoring the level of acetylcholine in the local interstitial fluid of the salivary glands is useful in detecting changes in parasympathetic nerve activity within the salivary glands.

In vivo, acetylcholine is rapidly degraded due to the activity of acetylcholinesterase. This enzyme is found in both central and peripheral nervous tissues. The effects of Eserine, an inhibitor of acetylcholinesterase, on acetylcholine concentrations in the dialysate were investigated using in vivo microdialysis of the central and peripheral tissues [21,26]. Perfusion with Eserine changed not only the basal level of acetylcholine induced by the inhibition of acetylcholinesterase, but also the relative acetylcholine output [26]. In the present study, acetylcholine concentration in the dialysate was maintained at a steady state for 360 min. This suggests that the Eserine concentration around the dialysis membrane would remain stable, even after 360 min. These findings are in good agreement with the results of an earlier fine structural cytochemistry analysis, which revealed high acetylcholinesterase activity in the intercellular and glandular stroma in rat submandibular glands [27]. Moreover, in the present study, the acetylcholine concentration in the dialysate returned to baseline levels after electrical stimulation, indicating that the effect of Eserine was transient and limited to the area around the dialysis membrane. In other words, these results show that Eserine does not denature the parasympathetic function of salivary glands.

Electrical stimulation of the *chorda tympani* nerve yielded a significant increase in the concentration of acetylcholine in the dialysate, to 28-fold higher than that at the basal level. This indicates that the acetylcholine levels in the dialysate mainly reflect that released from the nerve endings. These results strongly demonstrated that acetylcholine in the dialysate originates from the parasympathetic nervous system, which innervates the submandibular gland.

The present study results revealed the course of change in acetylcholine levels in the dialysate over a period of 360 min. The acetylcholine level in the dialysate was highly variable and unstable over the first 120 min after implantation of the probe, gradually decreasing and stabilizing thereafter. This time course is in good agreement with the results of earlier studies monitoring putative neurotransmitters in the brain [19,28,29].

Electrophysiological methods have been used to estimate changes in the activity of both the sympathetic and parasympathetic nerves innervating the submandibular gland [17]. This method is also useful in in vivo estimation of the net activity of the nervous system. Compared to this conventional method, microdialysis enables simultaneous monitoring of multiple bioactive substances in the interstitial fluid over time [30].

## 5. Conclusions

The results of the present study demonstrated that acetylcholine levels in the dialysate reflect acetylcholine levels in the interstitial fluid in the submandibular gland, and that an increase in acetylcholine level in the dialysate depends predominantly on the release of acetylcholine from the parasympathetic nerve endings. The microdialysis technique employed here offers a powerful tool for the monitoring of acetylcholine levels in the local interstitial fluids in salivary glands and for the detection of change in parasympathetic activity within the salivary glands.

## Figures and Tables

**Figure 1 biology-10-00351-f001:**
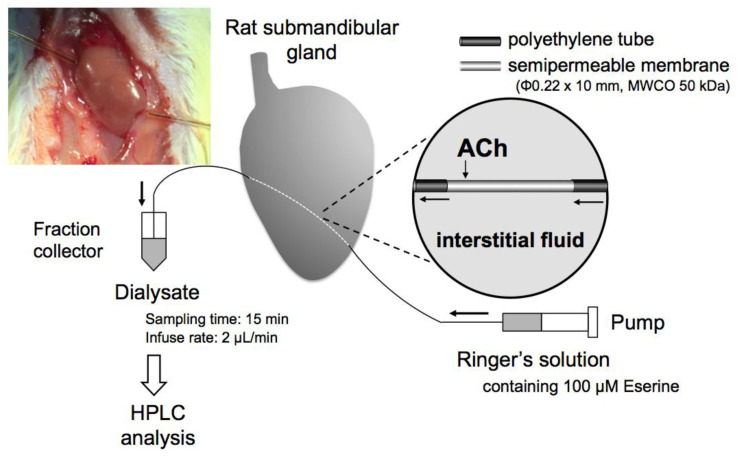
A diagram of the dialysis technique in the submandibular gland. The dialysis probes were perfused with Ringer’s solution containing Eserine (100 µM) using a micro-syringe pump.

**Figure 2 biology-10-00351-f002:**
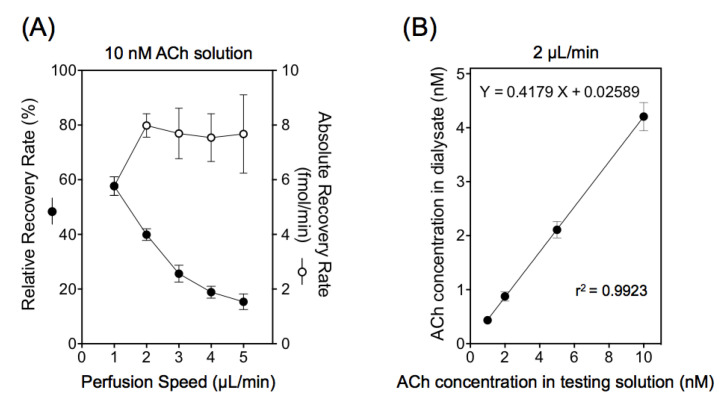
(**A**) The relationship between perfusion speed and relative and absolute recovery rates in vitro with 10 nM acetylcholine (ACh) in testing solution. The relative recovery rate = (concentration in dialysate)/(concentration in testing solution), and the absolute recovery rate = (concentration in dialysate) × (perfusion speed). (**B**) The relative recovery rate with different ACh concentrations in testing solution using a 2 µL/min perfusion speed. The values represent mean ± SD of 5 probes sampled for both A and B.

**Figure 3 biology-10-00351-f003:**
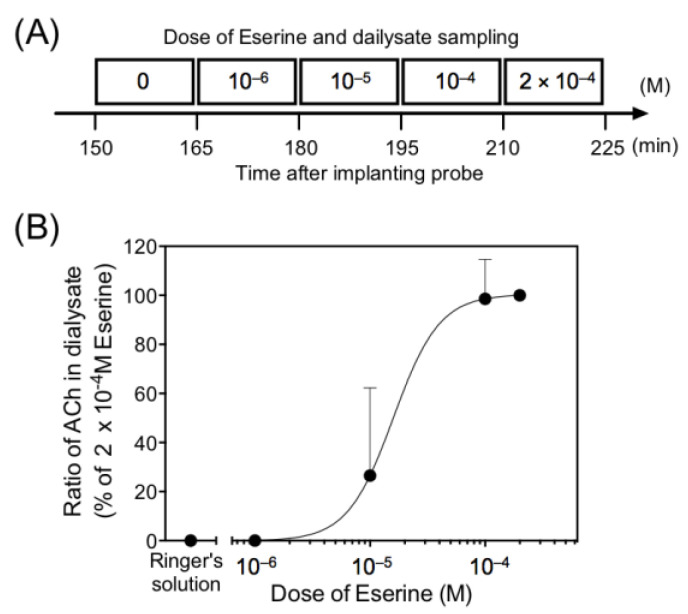
(**A**) A time course of sequential perfusion with different doses of Eserine and dialysate sampling. (**B**) The effect of Eserine perfusion on the extracellular concentration of acetylcholine (ACh) in rat submandibular glands. The values represent mean ± SD of 4 rats and are expressed as percentages of the acetylcholine levels obtained by perfusion with 2 × 10^−4^ M Eserine.

**Figure 4 biology-10-00351-f004:**
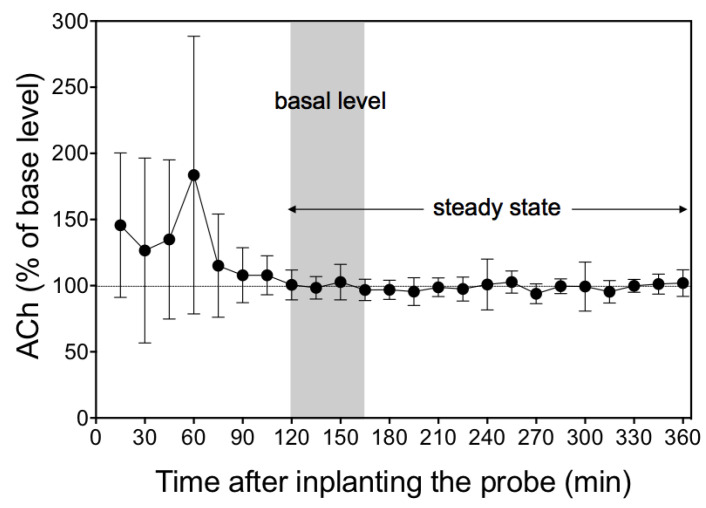
The time course of changes in acetylcholine (ACh) levels in the dialysate after probe implantation. The solid grey bar indicates 3 fractions for basal level. Acetylcholine concentration in the dialysate maintained an almost steady-state level for 240 min after probe implantation. The values represent mean ± SD of 6 rats and are expressed as percentages of the basal level.

**Figure 5 biology-10-00351-f005:**
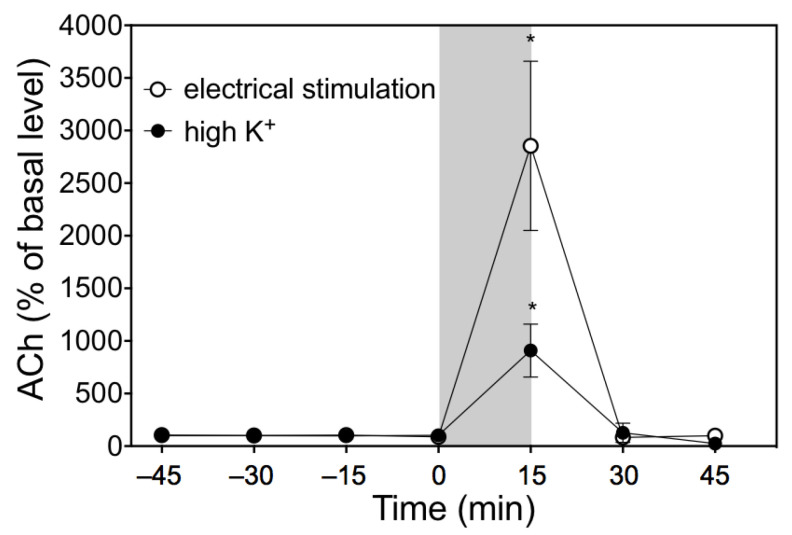
The effects of electrical stimulation of the *chorda tympani* nerve or perfusion with high-K^+^ Ringer’s solution on the acetylcholine (ACh) level in the dialysate. The solid grey bar indicates the length of electrical stimulation or perfusion with high-K^+^ Ringer’s solution. The values are means ± SD of 4 rats and are expressed as percentages of the basal level. Significantly different from the fraction at 0 min according to Dunn’s post-hoc test following Kruskal–Wallis test; * *p* < 0.05.

## Data Availability

The data that support the findings of this study are available from the corresponding author, [M.Y.], upon reasonable request.

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
