# Peer review of "In Vivo Monitoring of Acetylcholine Release from Nerve Endings in Salivary Gland"

_biology, 2021, doi:10.3390/biology10050351_

Round 1

Reviewer 1 Report

Review biology-118574 -v1

This is an interesting microdialysis study showing basal acetylcholine release and high potassium and electrically evoked acetylcholine (ACh) release in rat submandibular glands.  The paper may be improved by adding data on TTX-dependency and/or calcium-dependency of ACh release in salivary glands. It seems that ACh could only be detected in the presence of the acetylcholinesterase inhibitor physostigmine, which should be clearly stated.

Overall sound methods were used, but some experimental details need clarification. Some sentences or paragraphs in the main text need revision.

Abstract:  The reported level of basal ACh release in rat submandibular glands  was 4.8 nM , but – according to section 2.3 in Materials and Methods - this was in the presence of 100 µM physostigmine. If this is true, than this should be mentioned in the abstract as well.

Introduction

The aim of this (or future) work could be better defined, for instance to study interplay between parasympatic and sympatic nerve activity in salivary glands under various (patho)physiological conditions using microdialysis.

Lines 57-58: what underlying mechanism was demonstrated?

Materials and Methods

Section 2.3 and figure 3:

Was ACh below detection limit without physostigmine in the perfusion fluid? Please mention this explicitly, if this is the case.

How was the switch from regular Ringer’s to high-K+ Ringer’s and back to regular Ringer’s performed?

Was the sodium content adjusted in high-K+ Ringer’s? Please indicate.

How much IPHC was added to the samples? Please mention the amount.

Section 2.3 and figure 3:

What was the detection limit for ACh in the dialysate? Please mention this.

Results

Section 3.1 and figure 2:

Valued are means +/- SD of 5 dialysate samples. Do you mean 5 samples obtained with 5 different probes or 5 samples from 5 in vitro experiments with the same probe?  Please clarify.

Section 3.2 and figure 3:

  1. The protocol shows periods of 15 minutes with increasing doses of eserine. This protocol does not allow testing whether steady state levels of ACh are reached within 15 minutes.
  2. Figure B – x-axis: “Dose of eserine (mol)” should be “Dose of eserine (M)”

Figure B – y-axis: the ratio increases a  factor > 100 (from below 1% to 100%). Was the amount of added internal standard the same and could was the range of the ECD able to detect both endogenous ACh and added internal standard (in high doses used in the presence of 100 µM physostigmine)?

Section 3.3 and figure 4:

Indicate that basal levels were measured in the presence of 100 µM physostigmine  (if this is the case).

Discussion

line 215: “improved HPLC” – indicate what actually was improved as compared to earlier studies.

line 217: “ a variety of pathophysiological conditions in vivo” – give some examples for such conditions relevant for disturbances of parasympathetic (or sympathetic) activity in salivary glands.

Line 243: “effect of eserine was transient” – This is confusing. Please clarify what you mean.

Line 263: “advantage of microdialysis that it can be achieved using awake, freely moving animals” – This is true, but in this microdialysis study rats were anesthesized throughout the experiment (see section 2.3). Thus, using awake animals is something for future studies, or not? Clarify, or delete the comment on awake, freely moving animals.

Throughout the manuscript: minor corrections in English.

Author Response

This is an interesting microdialysis study showing basal acetylcholine release and high potassium and electrically evoked acetylcholine (ACh) release in rat submandibular glands.  

Response: We thank the reviewer for his/her kind remark.

The paper may be improved by adding data on TTX-dependency and/or calcium-dependency of ACh release in salivary glands.

Response: We thank the reviewer for raising this good point. We unfortunately did not investigate the effects of TTX and calcium on ACh release in salivary glands. We will be sure to do that in our future studies.

It seems that ACh could only be detected in the presence of the acetylcholinesterase inhibitor physostigmine, which should be clearly stated.

Response: We apologize for not being sufficient about this important matter in the original submission. We have changed and added the sentences in five places.

“The results revealed: 1) that mixing Eserine with Ringer’s solution allowed acetylcholine in the salivary glands to be quantified; 2) that acetylcholine concentrations in the dialysate were highly variable and unstable over the first 120 min after probe implantation, but reached a nearly stable level (4.8 ± 2.7 nM) thereafter in the presence of 100 µM Eserine;” (line 24-27: Abstract)

“No acetylcholine was detected in the dialysate in the absence of Eserine.” (page 6, line 16-17: section 3.2)

“Figure 4 shows the time course of change in acetylcholine level in the dialysate collected at 15-min intervals over a period of 360 min in the presence of 100 µM Eserine.” (page 6, line 23-25: section 3.3)

“Figure 5 shows changes in acetylcholine levels in the dialysate due to electrical stimulation of the chorda tympani nerve in the presence of 100 µM Eserine.” (page 6, line 30-32: section 3.4)

“Figure 5 shows changes in acetylcholine levels in the dialysate due to high-K+ Ringer’s solution in the presence of 100 µM Eserine.” (page 6, line 36-37: section 3.5)

Overall sound methods were used, but some experimental details need clarification. Some sentences or paragraphs in the main text need revision.

Response: We thank the reviewer for raising this point and have now included such a sentence in two places.

“The Ringer’s solution consisted of 147 mM NaCl, 2.2 mM CaCl2, 4.02 mM KCl, and the cholinesterase inhibitor Eserine (physostigmine; 100 µM). To elicit release of acetylcholine from nerve endings, the microdialysis probe was perfused with Ringer’s solution containing a higher than usual amount of KCI (100 nM)(high-K+ Ringer’s solution). The perfused solution was switched to high-K+ Ringer’s solution and back to regular Ringer’s solution by using an SI-60 liquid switch (Eicom).

   Each sampling period lasted 15 min (sample volume = 30 µl), which allows sufficient collection of effluent for a quantitative determination of acetylcholine level. Each 15-min sample was collected in a chilled microtube containing 3 µL of 3 nM IPHC as an internal standard.” (page 4, line 29-40: section 2.3)

“The present method achieved a detection limit of acetylcholine at 0.05 nM (1 fmol) and a quantitation limit of 0.2 nM (4 fmol) at a signal-to-noise ratio of 3. Quantification of the collected acetylcholine was evaluated with the IPHC peak area as the internal standard.” (page 5, line 14-18: section 2.4)

Abstract:  The reported level of basal ACh release in rat submandibular glands was 4.8 nM , but – according to section 2.3 in Materials and Methods - this was in the presence of 100 µM physostigmine. If this is true, than this should be mentioned in the abstract as well.

Response: We thank the reviewer for raising this point and have now included such a sentence in the text. Please see abstract.

“The results revealed: 1) that mixing Eserine with Ringer’s solution allowed acetylcholine in the salivary glands to be quantified; 2) that acetylcholine concentrations in the dialysate were highly variable and unstable over the first 120 min after probe implantation, but reached a nearly stable level (4.8 ± 2.7 nM) thereafter in the presence of 100 µM Eserine;“ (line 24-27: abstract)

Introduction: The aim of this (or future) work could be better defined, for instance to study interplay between parasympatic and sympatic nerve activity in salivary glands under various (patho)physiological conditions using microdialysis.

Response: We appreciate the reviewer’s suggestion and have now included such a sentence in the text. Please see introduction.

“The purpose of the present study was to evaluate parasympathetic activity in the salivary glands under various physiological conditions in vivo by using a microdialysis method to monitor acetylcholine released from nerve endings in the salivary glands of anesthetized rats.” (page 3, line 44 - page 4, line 3: introduction)

Lines 57-58: what underlying mechanism was demonstrated?

Response: We thank the reviewer for raising this point. We have corrected this statement. “These studies suggest that change in acetylcholine levels in homogenate from salivary glands results in modification of salivary secretory function.” (page 3, lines 24-25: introduction)

Materials and Methods

Section 2.3 and figure 3:

Was ACh below detection limit without physostigmine in the perfusion fluid? Please mention this explicitly, if this is the case.

Response: We thank the reviewer for raising this point and have now included such a sentence in the text.

“No acetylcholine was detected in the dialysate in the absence of Eserine.” (page 6, line 16-17: section 3.2)

How was the switch from regular Ringer’s to high-K+ Ringer’s and back to regular Ringer’s performed?

Response: We thank the reviewer for raising this point and have now included such a sentence in the text.

“The perfused solution was switched to high-K+ Ringer’s solution and back to regular Ringer’s solution by using an SI-60 liquid switch (Eicom).” (page 4, line 34-36: section 2.3)

Was the sodium content adjusted in high-K+ Ringer’s? Please indicate.

Response: We apologize for not being clear about this important matter in the original submission. We have now included such a sentence in the text.

“To elicit release of acetylcholine from nerve endings, the microdialysis probe was perfused with Ringer’s solution containing a higher than usual amount of KCI (147 mM NaCl, 2.2 mM CaCl2, 100 mM KCl, 100 µM Eserine) (high-K+ Ringer’s solution).” (page 4, line 31-34: section 2.3)

How much IPHC was added to the samples? Please mention the amount.

Response: We thank the reviewer for raising this point and have now included such a sentence in the text.

“Each sampling period lasted 15 min (sample volume = 30 µl), which allows sufficient collection of effluent for a quantitative determination of acetylcholine level. Each 15-min sample was collected in a chilled microtube containing 3 µL of 3 nM IPHC as an internal standard.” (page 4, line 37-40: section 2.3)

Section 2.3 and figure 3:

What was the detection limit for ACh in the dialysate? Please mention this.

Response: We thank the reviewer for raising this good point and have now included such a sentence in the text.

“The present method achieved a detection limit of acetylcholine at 0.05 nM (1 fmol) and a quantitation limit of 0.2 nM (4 fmol) at a signal-to-noise ratio of 3. Quantification of the collected acetylcholine was evaluated with the IPHC peak area as the internal standard.” (page 5, line 14-18: section 2.4)

Results

Section 3.1 and figure 2:

Valued are means +/- SD of 5 dialysate samples. Do you mean 5 samples obtained with 5 different probes or 5 samples from 5 in vitro experiments with the same probe?  Please clarify.

Response: We apologize for not being clear about this important matter in the original submission. We have corrected this statement in two places.

“Five dialysis probes were immersed in testing solution consisting of Ringer’s solution with a constant concentration of acetylcholine (10 nM; testing solution) at 37ºC and the dialysate samples were collected at vari-ous perfusion speeds (1–5 µl/min). The concentrations of acetylcholine in the dialysates obtained from the 5 different probes were measured by HPLC.” (page 6, line 1-5: section 3.1)

“Values represent mean ± SD of 5 probes sampled for both A and B.” (line 5 : legends for Fig 2)

Section 3.2 and figure 3:

  1. The protocol shows periods of 15 minutes with increasing doses of eserine. This protocol does not allow testing whether steady state levels of ACh are reached within 15 minutes.

Response: We thank the reviewer for raising this good point. The purpose of this experiment was to determine if Eserine is necessary for the detection of acetylcholine. We, in fact, are aware of the importance of equilibrium of Eserine concentration between the extracellular fluid and the perfusion solution. However, it would take too long to perfuse each concentration of eserine for more than 15 minutes each to maintain equilibrium in the same individual. We will be sure to do that in our future studies.

  1. Figure 3B – x-axis: “Dose of eserine (mol)” should be “Dose of eserine (M)

Response: We thank the reviewer for raising this point, and correcting us. We have now changed x-axis in Figure 3B. Please see Figure 3B.

Figure B – y-axis: the ratio increases a  factor > 100 (from below 1% to 100%). Was the amount of added internal standard the same and could was the range of the ECD able to detect both endogenous ACh and added internal standard (in high doses used in the presence of 100 µM physostigmine)?

Response: We thank the reviewer for raising this point. The data was expressed as percentages of level of 2x10-4M Eserine. The amount of the added internal standard is the same, and we are able to detect both endogenous ACh and the added internal standard within the detection range of the ECD.

Section 3.3 and figure 4:

Indicate that basal levels were measured in the presence of 100 µM physostigmine  (if this is the case).

Response: We thank the reviewer for raising this point and have now included such a sentence in the text.

“The acetylcholine levels were highly variable and unstable over the first 120 min after implantation of the probe, but showed a gradual decrease to basal level (4.9 ± 2.2 nM) and a stable state (4.8±2.7 nM) thereafter.” (page 6, line 25-28: section 3.2)

Discussion

line 215: “improved HPLC” – indicate what actually was improved as compared to earlier studies.

Response: We apologize for not being clear about this important matter in the original submission. Based on the reviewer’s comments, we changed word from “improved HPLC” to “highly sensitive HPLC”. (page 9, line 26)

line 217: “ a variety of pathophysiological conditions in vivo” – give some examples for such conditions relevant for disturbances of parasympathetic (or sympathetic) activity in salivary glands.

Response: We thank the reviewer for raising this point and have now included such a sentence in the text.

“For example, while saliva flow showed a marked decrease in streptozotocin-induced diabetic mice, acetylcholine in homogenates obtained from their salivary glands showed a significant increase [12]. Cellular mechanisms such as the synthesis of acetylcholine and compensatory changes due to diabetes have failed to explain this discrepancy.” (page 9, line 29 - page 10, line 3)

Line 243: “effect of eserine was transient” – This is confusing. Please clarify what you mean.

Response: We apologize for not being clear about this important matter in the original submission and have now included a sentence in the text.

“In other words, these results show that Eserine does not denature the parasympathetic function of salivary glands. (page 10, line 38 - 40)

Line 263: “advantage of microdialysis that it can be achieved using awake, freely moving animals” – This is true, but in this microdialysis study rats were anesthesized throughout the experiment (see section 2.3). Thus, using awake animals is something for future studies, or not? Clarify, or delete the comment on awake, freely moving animals.

Response: We thank the reviewer for raising this point and have now deleted the comment on awake, freely moving animals in the text.

Throughout the manuscript: minor corrections in English.

Response: Prof. Jeremy David Williams, Department of International Medical Communications Tokyo Medical University, edited the English text of the manuscripts.

Reviewer 2 Report

Yoshikawa and Kawaguchi present an examination of the possible in vivo monitoring of acetylcholine release from nerve ending in salivary glands

The authors demonstrated that the microdialysis technique can be a powerful tool for monitoring of acetylcholine levels in local interstitial fluids of salivary glands and detection of changes in parasympathetic activity within the salivary glands.

The study is largely well-controlled and appropriately interpreted, as well the data are presented clearly. A few suggestions for improvement are listed below.

Minor points:

1)Line 70: For better readability and clarity please specify in the introduction that the microdialysis technique was applied in anesthetized rats (no freely moving animals).

2) The authors in the discussion section reported that “Compared to conventional methods, microdialysis enables simultaneous monitoring  several bioactive substances in the interstitial fluid over time. In addition, the great advantage of microdialysis is that it can be achieved using awake, freely moving animals”. However, they performed the microdialysis in anesthetized rats (no freely moving animals). I think it would be helpful to compare and contrast the differences due to this different approaches (anesthetized vs freely moving), as well as to discuss the feasibility of this latter approach, in the discussion section.  

Author Response

The authors demonstrated that the microdialysis technique can be a powerful tool for monitoring of acetylcholine levels in local interstitial fluids of salivary glands and detection of changes in parasympathetic activity within the salivary glands.

The study is largely well-controlled and appropriately interpreted, as well the data are presented clearly. A few suggestions for improvement are listed below.

Response: We thank the reviewer for his/her kind remark.

1)Line 70: For better readability and clarity please specify in the introduction that the microdialysis technique was applied in anesthetized rats (no freely moving animals).

Response: We thank the reviewer for raising this point and have now included such a sentence in Introduction.

“The purpose of the present study was to evaluate parasympathetic activity in the salivary glands under various physiological conditions in vivo by using a microdialysis method to monitor acetylcholine released from nerve endings in the salivary glands of anesthetized rats.” (page 1, line 44 - page 2, line 3)

2) The authors in the discussion section reported that “Compared to conventional methods, microdialysis enables simultaneous monitoring  several bioactive substances in the interstitial fluid over time. In addition, the great advantage of microdialysis is that it can be achieved using awake, freely moving animals”. However, they performed the microdialysis in anesthetized rats (no freely moving animals). I think it would be helpful to compare and contrast the differences due to this different approaches (anesthetized vs freely moving), as well as to discuss the feasibility of this latter approach, in the discussion section.  

Response: We thank the reviewer for raising this good point. In this microdialysis study, rats were anesthesized throughout the experiment. We will be sure to do that in our future studies. We deleted the sentences about awake, freely moving animals in revised manuscript.

Reviewer 3 Report

The authors have applied a microdialysis method for the detection of acetylcholine level in salivary gland in vivo. Thus, the work is really interesting; however, there are several minor issues that need our attention.

1. The Introduction in its present form is pretty clumsy. Therefore, it
should be rewritten, especially see the lines 50-61.
Moreover, revise unclear statements: "leads to of noradrenaline" (line 39),
"that contents of neurotransmitters in homogenate" (line 53), and also
"the microdialysis dialysis" (line 215).
2. A number of animals used for the experiments should be mentioned.
See the Chapter 2.1 and the figure legends.

Author Response

The authors have applied a microdialysis method for the detection of acetylcholine level in salivary gland in vivo. Thus, the work is really interesting; however, there are several minor issues that need our attention.

Response: We thank the reviewer for his/her kind remark.

  1. The Introduction in its present form is pretty clumsy. Therefore, it
    should be rewritten, especially see the lines 50-61.

Response: We thank the reviewer for raising this point and rewrote the Introduction. Please see introduction.

lines 50-60:

from “The acute control of salivary secretion was demonstrated using animal models under anesthesia. Perfusion with parasympathomimetics into salivary glands or electrical stimulation of nerves significantly increases salivary flow. Previous study demonstrated that contents of neurotransmitters in homogenate from salivary glands of rats or mice. The concentrations of acetylcholine in sublingual of rats were higher than those of other glands. Chronic administration of isoprenaline (non-selective β-adrenergic agonist) or streptozotocin (diabetogenic drug) altered the contents of acetylcholine in homogenates of the salivary glands. These studies demonstrated the mechanism underlying actions of acetylcholine on salivation by measuring its concentrations in homogenates from salivary glands. However, findings obtained by these studies do not always correspond to the physiological and functional significances of acetylcholine levels in the salivary glands.”

to

“Previous studies have demonstrated neurotransmitter content in homogenate obtained from rat and mouse salivary gland [9, 10]. Chronic administration of isoprenaline, a non-selective β-adrenergic agonist, or streptozotocin, a diabetogenic drug, altered both the amount of saliva and acetylcholine content in homogenate obtained from salivary gland [11, 12]. These studies suggest that change in acetylcholine levels in homogenate from salivary glands results in modification of salivary secretory function. Acetylcholine in homogenate, however, includes that released in interstitial fluid, as well as that stored in the cells. The activity of the parasympathetic nervous system is mainly determined by the amount of acetylcholine released. This is supported by the results of earlier studies in which the salivary glands were perfused with parasympathomimetics [13-16] or in which electrical stimulation of the nerves resulted in a significant increase in salivary flow [17]. This indicates that it is necessary to measure the amount of acetylcholine released in order to determine parasympathetic activity in the salivary glands.”

Moreover, revise unclear statements: "leads to of noradrenaline" (line 39),
"that contents of neurotransmitters in homogenate" (line 53), and also
"the microdialysis dialysis" (line 215).

Response: We thank the reviewer for raising this point, and correcting us.

line 39: “On the other hand, sympathetic excitation leads to of noradrenaline from postganglionic nerve endings.” (page 3, line 7-9)

to

“Sympathetic excitation, on the other hand, leads to the release of noradrenaline from postganglionic nerve endings.”

line 53: “Previous study demonstrated that contents of neurotransmitters in homogenate from salivary glands of rats or mice.”

to

“Previous studies have demonstrated neurotransmitter content in homogenate obtained from rat and mouse salivary gland.” (page 3, line 19-20)

line 215: “the microdialysis dialysis” to “microdialysis” (page 9, line 26)

  1. A number of animals used for the experiments should be mentioned.
    See the Chapter 2.1 and the figure legends.

Response: We thank the reviewer for raising this point and have now included such a sentence in 4 places.

“Male Wistar rats (8-9 weeks-old, 230–280 g each, n=18; Nihon Clea, Tokyo, Japan) were housed in an air-conditioned room at a control temperature of 24ºC–26ºC and 50%–60% humidity, with a 12-hour light/dark cycle (lights on: 07:00 hrs) and food and water freely available.” (page 4, line 11-14, Chapter 2.1).

“Values represent mean ± SD of 4 rats and expressed as percentages of level of 2x10-4 M Eserine.” (Legends for Figure 3)

“Values represent mean ± SD of 6 rats and are expressed as percentages of basal level.” (Legends for Figure 4)

“Values are means ± SD of 4 rats and are expressed as percentages of basal level.” (Legends for Figure 5)